# Alteration of Bacterial Communities in Anterior Nares and Skin Sites of Patients Undergoing Arthroplasty Surgery: Analysis by 16S rRNA and Staphylococcal-Specific *tuf* Gene Sequencing

**DOI:** 10.3390/microorganisms8121977

**Published:** 2020-12-12

**Authors:** Søren Iversen, Thor Bech Johannesen, Anna Cäcilia Ingham, Sofie Marie Edslev, Staffan Tevell, Emeli Månsson, Åsa Nilsdotter-Augustinsson, Bo Söderquist, Marc Stegger, Paal Skytt Andersen

**Affiliations:** 1Department of Bacteria, Parasites and Fungi, Statens Serum Institut, 2300 Copenhagen, Denmark; thej@ssi.dk (T.B.J.); anmc@ssi.dk (A.C.I.); smed@ssi.dk (S.M.E.); mtg@ssi.dk (M.S.); psa@ssi.dk (P.S.A.); 2Department of Infectious Diseases and Centre for Clinical Research and Education, County Council of Värmland, SE-651 82 Karlstad, Sweden; staffan.tevell@regionvarmland.se; 3School of Medical Sciences, Faculty of Medicine and Health, Örebro University, SE-701 82 Örebro, Sweden; emeli.mansson@regionvastmanland.se (E.M.); bo.soderquist@oru.se (B.S.); 4Centre for Clinical Research, Hospital Västmanland, Uppsala University, Region Västmanland, SE‐721 89 Västerås, Sweden; 5Department of Infectious Diseases and Department of Clinical and Biomedical Sciences, Linköping University, SE-581 83 Norrköping, Sweden; asa.nilsdotter-augustinsson@liu.se; 6Department of Laboratory Medicine, Clinical Microbiology, Faculty of Medicine and Health, Örebro University, SE-701 82 Örebro, Sweden

**Keywords:** arthroplasty, prosthetic joint replacement, prosthetic joint infection, systemic antibiotic prophylaxis, antibiotics, chlorhexidine gluconate, coagulase-negative staphylococci, *tuf* gene sequencing, staphylome, 16S rRNA gene sequencing, microbiome

## Abstract

The aim was to study alterations of bacterial communities in patients undergoing hip or knee arthroplasty to assess the impact of chlorhexidine gluconate soap decolonisation and systemic antibiotic prophylaxis. A Swedish multicentre, prospective collection of samples obtained from elective arthroplasty patients (n = 83) by swabbing anterior nares, skin sites in the groin and the site of planned surgery, before and after arthroplasty surgery, was analysed by 16S rRNA (V3-V4) gene sequencing and a complementary targeted *tuf* gene sequencing approach to comprehensively characterise alterations in staphylococcal communities. Significant reductions in alpha diversity was detected for both bacterial (*p* = 0.04) and staphylococcal (*p* = 0.03) groin communities after arthroplasty surgery with significant reductions in relative *Corynebacterium* (*p* = 0.001) abundance and *Staphylococcus hominis* (*p =* 0.01) relative staphylococcal abundance. In nares, significant reductions occurred for *Staphylococcus hominis* (*p* = 0.02), *Staphylococcus haemolyticus* (*p* = 0.02)*,* and *Staphylococcus pasteuri* (*p* = 0.003) relative to other staphylococci. *Staphylococcus aureus* colonised 35% of anterior nares before and 26% after arthroplasty surgery. *Staphylococcus epidermidis* was the most abundant staphylococcal species at all sampling sites. No bacterial genus or staphylococcal species increased significantly after arthroplasty surgery. Application of a targeted *tuf* gene sequencing approach provided auxiliary staphylococcal community profiles and allowed species-level characterisation directly from low biomass clinical samples.

## 1. Introduction

Arthroplasties are common surgical procedures performed to alleviate pain and to restore function in damaged or worn-out joints. Colonising opportunistic pathogens (COPs) constitute part of the normal human microbiota [1] and comprise a risk for establishment of a prosthetic joint infection (PJI) [2]. Surgical procedures that entail breaching the epidermal protective layer increase the risk of invasive infections and an estimated 70–95% of all surgical site infections (SSIs) arise from a COP residing in the microbiota of a patient [3]. For arthroplasty, PJI incidence rates vary from low in hips, knees and shoulders (<2%), to higher risk in elbows (2–11%) and ankles (2–20%) [4]. *Staphylococcus aureus* and coagulase-negative staphylococci (CoNS) collectively account for more than half of all causative organisms derived from PJIs, with *S. aureus* being overrepresented in early postinterventional and in late acute hematogenous PJIs, and CoNS in chronic PJIs and more often as part of polymicrobial infections [2,5,6,7]. Recent methodological advances using matrix-assisted laser desorption ionisation–time of flight (MALDI-TOF) mass spectrometry (MS) have established *Staphylococcus epidermidis* as the most frequently isolated CoNS in PJIs (60%) followed by *Staphylococcus capitis* (11%), *Staphylococcus lugdunensis* (10%), *Staphylococcus warneri* (4%), *Staphylococcus hominis* (3%), and *Staphylococcus haemolyticus* (3%) [8,9]. In hip and knee arthroplasty, non-staphylococcal PJIs are most commonly caused by aerobic Gram-negative rods (9%), particularly *Escherichia coli* and *Pseudomonas aeruginosa,* followed by streptococci (8%), enterococci (3%), and anaerobic Gram-positive rods (4%)—typically *Cutibacterium acne*s [2].

Microbial communities vary widely across different body sites and specific bacterial genera are associated with physiological differences of the skin [10,11]. The genera *Corynebacterium* and *Staphylococcus* are integral members of a healthy human microbiota and are commonly present on moist skin sites including the groin (inguinal crease) and the anterior nasal cavities [12,13,14]. In the latter, seven distinct nasal community state types (CSTs) have been described, where each CST is characterised by a predominant bacterial genus or species, e.g., *S. aureus* and *S. epidermidis* [15]. The *Staphylococcus* genus has undergone extensive evolutionary adaptation to human and animal hosts and exhibits a predominantly asymptomatic and commensal lifestyle [16]. However, the ubiquity of staphylococci, combined with their efficient acquisition of antimicrobial resistance genes and ability to form biofilm, makes staphylococcal COPs a primary concern from a clinical perspective, especially with the documented spread of multidrug-resistant (MDR) *S. epidermidis* lineages [12,17]. To reduce the risk of developing a PJI, patients scheduled for arthroplasty routinely undergo mandatory preoperative decolonisation procedures that, according to Swedish guidelines (https://lof.se/patientsakerhet/vara-projekt/rekommendationer/), include preoperative showers with chlorhexidine gluconate (CHG) soap along with pre- and perioperative prophylaxis with systemic antibiotics to lower the total bacterial burden [18]. However, the impact of these treatments on the human microbiota are complex and needs better understanding.

In this study, we analysed swabs from anterior nares as well as skin swabs from groin and the site planned for surgery of patients undergoing hip or knee arthroplasty, to study bacterial communities before and after arthroplasty surgery and preoperative decolonisation treatment with CHG soap showers and systemic antibiotic prophylaxis. To characterise the effects on *Staphylococcus* communities in detail, a novel amplicon-based *Staphylococcus*-centric “staphylome” approach was developed that targets the conserved and essential *tuf* gene, encoding for Elongation Factor Thermal Unstable (EF-Tu). This approach provides staphylococcal species differentiation and direct applicability to low abundant and polymicrobial clinical samples, with greater resolution than 16S rRNA gene sequencing [19,20].

## 2. Materials and Methods

### 2.1. Sample Collection

A prospective study was performed at the Departments of Orthopaedics at the University Hospitals of Örebro and Linköping (Sweden), and the County hospitals of Karlstad and Västmanland (Sweden) from 2013 to 2015. Patients (n = 83) scheduled for arthroplasty surgery were included in the study. Swabbing (Eswab, Copan Italia S.p.A., Brescia, Italy) was performed in a standardised manner (i) in anterior nares (ii) on the skin of the ipsilateral groin, and (iii) on the skin area planned for surgery (hip or knee, hereafter referred to as operation site). Sampling was performed preoperatively at the outpatient clinic visit 2–6 weeks scheduled arthroplasty (before) and postoperatively just before discharge from the hospital (after). The Eswab transport medium was stored at −80 °C pending further analysis. All patients underwent two preoperative showers using CHG-containing soap and the site of surgery was disinfected with a 0.5% chlorhexidine solution in 70% ethanol in the operating theatre by the scrub nurse. Antimicrobial prophylaxis was administered (i.e., for non-penicillin allergic patients, 2 g cloxacillin 30–45 min before skin incision followed by two additional doses after 2 and 6 h, respectively) as per clinical routine in Sweden.

### 2.2. 16S rRNA Gene Sequencing and Sequence Preprocessing

Extraction of DNA was done using FastDNA SPIN kit for Soil kits (MP Biomedicals, LLC, Solon, OH, USA) according to the manufacturer’s protocol. Bacterial community profiling was performed using universal 16S rRNA V3-V4 gene primers as previously described [21] and sequences were classified using BION-meta [22] performing sequence cleaning, dereplication, chimera filtering and calculating similarities between sequences before matching against the Ribosomal Protein Database (RDP) [23]. The configuration of the software was adjusted to match the primers used and run with default parameters [22]. We initially detected high levels of contaminating *Burkholderiaceae*, in particular *Ralstonia solcenarium* and *Burkholderia* spp. However, these sequences were also detected in negative extraction controls and have been described as kit contaminants in the extraction kits that were used for DNA extraction [24]. Hence, we applied the prevalence method (threshold 0.5) of the R package decontam (version 1.4.0) to filter out contaminant sequences associated with these and other taxa [25]. Furthermore, we manually removed additional contaminants prior to downstream analysis (see details in https://github.com/ssi-dk/staphylome/tree/master/scripts).

### 2.3. Development of a New tuf Gene Sequencing “Staphylome” Approach

We retrieved whole genome data from all publicly available staphylococci (>9000 genomes accessed 24 July 2018) from RefSeq (https://www.ncbi.nlm.nih.gov/refseq/) to generate a staphylococcal reference database. From this database, *tuf* gene sequences were extracted and aligned to select suitable primer-binding sites. The staphylococcal *tuf* gene database is available at (https://github.com/ssi-dk/staphylome/tree/master/database). Amplification of a 491 bp region in the *tuf* gene was performed using primers with heterogeneity spacers (Appendix A) in PCR reactions using KAPA HiFi HotStart ReadyMix (2x) Polymerase in 25 μL reaction volumes [12.5 μL 2xKAPA, 0.2 μm forward and reverse primer, 10 μL DNA template] with the PCR program (3 min at 95 °C, 25 cycles: 98 °C for 20 s, 60 °C for 15 s, 72 °C for 45 s, and final extension 72 °C for 5 min). Indexing of amplicons was done using Illumina’s Nextera XT indexing kits (Illumina Inc., San Diego, CA, USA) in 25 μL reaction volumes [12.5 μL 2xKAPA HiFi, 2.5 μL forward and reverse index primers, and 2 μL DNA template] with the PCR program (3 min at 95 °C, 20 cycles of: 98 °C for 30 s, 55 °C for 30 s, 72 °C for 45 s, and a final extension at 72 °C for 5 min). After Agencourt AMPure XP (Beckman Coulter, Inc. Indianapolis, IN, USA) bead purification (x0.8 ratio), normalisation and pooling, libraries were sequenced on Illumina MiSeq (Illumina Inc., San Diego, CA, USA) with a 600-cycle V3 kit (Illumina Inc., San Diego, CA, USA) with 10% PhiX spiked in.

Demultiplexing of raw *tuf* gene reads was done using the bcl2fastq Conversion Software (Illumina) to obtain one forward and one reverse FASTQ file for each sample. The cutadapt software (version 2.3) was used to trim off heterogeneity spacers and primers at a tolerated maximum error rate of 8% corresponding to one mismatch [26]. Both read pairs were discarded if a primer was not found in one of the reads. The R package DADA2 (version 1.12.1) was used to infer amplicon sequence variants (ASVs) at single-nucleotide resolution [27]. We truncated forward reads at 270 bp and reverse reads at 241 bp to obtain reads with 20 bp overlap for merging and to retain sufficient read quality. The DADA2 pipeline (version 1.12.1) was executed with default settings (except truncation lengths) per sequencing run. The resulting ASV tables were then combined and chimeras were identified per sample, but removed from all samples globally (*removeBimeraDenovo*() function, method “consensus”). We formatted our staphylococcal *tuf* gene database to be compatible with DADA2’s *assignTaxonomy*() function and used it to classify the staphylococcal ASVs with the RDP naive Bayesian classifier method. Unclassified reads were excluded from downstream analysis.

### 2.4. Statistical Analysis

All statistical analyses were performed in R (version 3.6.0, R Foundation for Statistical Computing, Vienna, Austria) [28]. Amplicon sequence variant (ASV) count tables, taxonomy tables and related metadata (data sets available at https://github.com/ssi-dk/staphylome/tree/master/data) were integrated with the R package phyloseq and its dependencies [29]. Packages used for the generation of plots include ggplot2 (version 3.2.1) and gplots [30,31]. Based on rarefaction curves we determined a lower cut-off at 2000 reads per sample for both 16S rRNA (V3-V4) and *tuf* samples and excluded sample pairs from downstream analysis if one sample was below this cutoff (Appendix A). Ordination analysis by principal coordinates analysis (PCoA) based on Bray–Curtis dissimilarity was performed on Hellinger-transformed count data for both 16S rRNA (V3-V4) and *tuf* samples. We used the *betadisper*() function to test whether the variances of the groups we compared were homogenous and tested for differences in community structure between body sites and time points with permutational multivariate analysis of variance (PERMANOVA) using distance matrices with the *adonis*() function from the vegan package (version 2.5.5) [32]. Bacterial alpha diversity by means of the Shannon index was calculated on untransformed 16S rRNA (V3-V4) read counts and staphylococcal *tuf* gene ASV counts, respectively, and alpha diversity before and after antimicrobial prophylactic treatment and arthroplasty was compared by paired Wilcoxon signed-rank tests. The 16S rRNA (V3-V4) read counts and staphylococcal *tuf* gene ASV counts were agglomerated at genus and species level, respectively, and transformed to relative abundance and visualised as bar plots with the top 10 most abundant bacterial genera and staphylococcal species displayed for each body site. We calculated the patient-wise change in percentage points from before to after arthroplasty for the most abundant bacterial genera and staphylococcal species and used paired Wilcoxon signed-rank tests with subsequent Benjamini–Hochberg correction for multiple testing to determine which bacterial genera and staphylococcal species changed significantly after decolonisation treatment and arthroplasty. Scripts for these analysis steps are available at (https://github.com/ssi-dk/staphylome/tree/master/scripts).

### 2.5. Ethical Approval

The collection of pre- and postoperative samples swabs was approved by the Regional Ethical Review Board of Uppsala (reference: 2012/092 and amendment 2020-00991).

## 3. Results

### 3.1. Bacterial Community Alterations Assessed by 16S rRNA (V3-V4) Gene Sequencing

Alteration of bacterial communities in anterior nares, groin, and operation sites of hip or knee arthroplasty patients, sampled before preoperative decolonisation treatments and after arthroplasty surgery, was first examined by use of 16S rRNA (V3-V4) gene sequencing. A large proportion of the bacterial communities were not included in the analysis as too few reads (< 2000) remained after contaminant filtering, especially for samples obtained from groin and operation sites following decolonisation treatment and arthroplasty surgery (Appendix A and Appendix A).

PCoA analysis showed that bacterial communities clustered according to body site (Figure 1A) and PERMANOVA revealed no significant difference (*p* = 0.2) in nasal bacterial community structure before compared to after decolonisation treatment and arthroplasty surgery (Figure 1B). Antibiotic prophylaxis and CHG decolonisation were associated with small nonsignificant reductions in alpha diversity in anterior nares and significant reductions in alpha diversity at groin sites (*p* = 0.04) (Appendix A). Among the top 10 bacterial genera in nares, only *Dolosigranulum* displayed a significant decrease across the patient cohort (*p* = 0.002) (Figure 1D). In the groin, most patients were colonised by *Staphylococcus, Corynebacterium*, and *Anaerococcus* prior to decolonisation treatments and arthroplasty (Figure 1C), and a significant reduction was detected for *Corynebacterium* (*p* = 0.0014) following arthroplasty (Figure 1E). Significantly different within-group variances between bacterial groin communities before compared to after arthroplasty (*betadisper*, *p* = 0.01) likely resulted in significantly altered groin communities after arthroplasty (PERMANOVA, *p* = 0.006). The *Staphylococcus* genus showed small, nonsignificant reductions in nares and groin, with individual patients experiencing either relative increase or decrease (Figure 1D,E). In groin sites, we also noted a trend with increased colonisation by Gram-negative bacteria such as *Escherichia* and *Klebsiella* (Figure 1D).

### 3.2. Development and Validation of the Staphylome Approach

To explore low-abundant *Staphylococcus* communities in greater detail, we developed and validated a *Staphylococcus*-specific approach capable of detecting and differentiating individual *Staphylococcus* species. The vast majority of staphylococcal *tuf* genes cluster according to species as determined from whole genome sequencing (WGS) data, with only *Staphylococcus cohnii* being represented in two distinct clades (Figure 2A). We designed a new set of *tuf* gene primers from this dataset and validated these primers on 23 different staphylococcal species and on common skin and nasal isolates (Appendix A). Next, we generated staphylococcal community profiles in triplicates from a mock community comprised of equal amounts of genomic DNA from six different *Staphylococcus* species. A dilution series revealed that all six *Staphylococcus* species could be detected down to ≈10 copies/organism with decreasing technical reproducibility in the low-abundant mock communities (Figure 2B). Dilution down to ≈1 copy/organism resulted in detection of between two and four of the six species in each triplicate.

### 3.3. Staphylococcal Community Alterations Assessed by the Staphylome Approach

We applied the staphylome approach to study subtle alterations in staphylococcal community composition in anterior nares, groin, and operation site samples before and after arthroplasty surgery. Most patients colonised with *S. aureus* in the nares (23 out of 65) before prophylactic treatments (here, colonisation signifies *S. aureus* ASVs ≥1% of total *tuf* sequencing reads) retained *S. aureus* colonisation after arthroplasty (17 out of 23) and one patient gained *S. aureus* without being colonised prior to arthroplasty surgery.

We found a positive correlation (*p* = 0.04, *ρ* = 0.36) between *S. aureus* and total staphylococcal relative abundance in the nares, where increased presence of *S. aureus* for three patients—P4, P41, and P43—coincided with increased total staphylococcal relative abundance (Appendix A). Three staphylococcal species, *S. hominis* (*p* = 0.02), *S. haemolyticus* (*p* = 0.02)*,* and *S. pasteuri* (*p* = 0.003), showed a significant relative decrease in the nares after arthroplasty surgery across the patient cohort (Figure 3, Appendix A). The most abundant staphylococcal species, both before and after arthroplasty, at all sampling sites was *S. epidermidis,* that showed small nonsignificant increases in relative staphylococcal abundance at all sampling sites after arthroplasty surgery, together with *S. lugdunensis* and *S. capitis*. However, the increase in these CoNS did not correlate with increased total staphylococcal relative abundance (Appendix A). Staphylococcal community alterations in groin and operation sites clustered into two distinct groups characterised by alterations in *S. epidermidis* relative abundance (Figure 3E,F). In groin samples, alpha diversity among *Staphylococcus* species decreased significantly (*p* = 0.03) together with a significant reduction in *S. hominis* (*p =* 0.01) (Figure 3B,E and Appendix A). In operation sites we detected a significant reduction in *S. haemolyticus* (*p* = 0.039) after decolonisation treatment and arthroplasty (Figure 3C,F).

## 4. Discussion

The data presented here suggest that systemic antibiotic prophylaxis and decolonisation treatment with CHG soap in association with prosthetic joint replacement surgery and hospitalisation can alter bacterial communities with variations across different body sites. Similar alterations were observed for subsets of patients, while other alterations were patient specific. Overall, bacterial and staphylococcal communities remained stable in the nares after decolonisation and arthroplasty, albeit with small nonsignificant reductions in alpha diversity and significant decreases in *Dolosigranulum* relative abundance and significant decrease in *S. hominis*, *S. haemolyticus,* and *S. pasteuri* relative staphylococcal abundance. Groin communities had significant reductions in alpha diversity in both bacterial and staphylococcal communities after arthroplasty with significantly less *Corynebacterium* and *S. hominis*. The patient specific alterations observed between individuals are strongly influenced by the distinctive microbiomes present prior to arthroplasty. Since the human nares are known to sustain highly individual bacterial communities, the alterations observed are therefore strongly influenced by the antecedent bacterial community of each patient [15,33]. The differences observed between sampling sites similarly reflect differences in bacterial communities at each site, but also differential exposure to CHG decolonisation treatment, since the nares, receiving no topical intranasal decolonisation, were predominately affected by systemic antibiotic prophylaxis, whereas skin sites were affected by both systemic antibiotic prophylaxis, preoperative CHG soap showers, and perioperative skin disinfection at operation sites. In this study, a lower success rate was achieved when analysing groin and operation site samples following decolonisation treatments and arthroplasty for both 16S rRNA (V3-V4) gene sequencing and the staphylome approach, which points to an intended efficacy of the decolonisation treatments (Appendix A and Appendix A). Future studies incorporating absolute abundance data on the studied microbial communities as well as nontreated control subjects could help to clarify the nature of the observed alterations and assist in determining if the changes arise from preoperative decolonisation treatment, antibiotic prophylaxis, changes in environment due to hospitalisation, common fluctuations in the bacterial composition, or a combination thereof. Our results are in agreement with previous culture-based findings stating that CHG showers do not eradicate bacteria, but instead decrease bacterial diversity [34]. A *tuf* gene-based study similarly reported that nasal staphylococcal communities were not substantially altered after antibiotic treatment [35]. Our results are in general agreement with previous culture-based findings [33] in regard to bacterial (Appendix A) and staphylococcal species prevalence (Appendix A), particularly after a filtering approach is applied to exclude low abundant species present ≤1% of total reads in each sample. Indeed, the observed prevalence is severely affected by application of filtering, which should be investigated further moving forward with amplicon-based approaches, especially as more studies describe confounding effects of sequencing-based methods e.g., index-hopping and sample cross-talk [36]. In general, comparative studies of culture-based and amplicon-based approaches are needed, e.g., using live microbial mock communities, to clarify if discrepancies arise due to limitations in culturing or sequencing methodology, and potentially establish limitations to both approaches. The sensitivity of amplicon-based methods allows for analysis of low abundant bacterial communities, but simultaneously it increases the risk of contamination from the laboratory environment [37]. Contamination issues also impacted this study, including DNA extraction kit contamination issues previously reported by Salter et al. 2014 [24]. Application of the staphylome approach was not impacted by systematic contamination as staphylococci are not common kit contaminants in the extraction kits that were used, but sporadic staphylococcal contamination did occur in negative controls, likely due to cross-contamination between samples. However, since these contaminating *tuf* gene ASVs overlap with the most abundant staphylococcal ASVs present in the samples, a decontamination approach was not pursued. When swabbing anterior nares there is a risk of contamination from adjacent skin sites and some bacterial species might be underrepresented compared to other studies as e.g., *S. aureus* is reported to increase in prevalence at other sites like in the posterior vestibule [33].

## 5. Conclusions

Our data support the use of amplicon-based sequencing approaches for studying alterations in low abundant bacterial communities and highlight the use of a *Staphylococcus* genus-centric approach targeting the *tuf* gene as it contains sufficient sequence diversity to distinguish between closely related staphylococcal species, while enabling PCR amplicon products of a suitable size (491 bp) for multiplexed sequencing on the Illumina MiSeq platform (Figure 2 and Appendix A). This staphylome approach could also be applied when examining subtle dynamics of the skin microbiota in other study settings, e.g., patients with atopic dermatitis or haematological malignancies, patients undergoing chemotherapy or in neonates. It remains an open question how current standard preoperative prophylactic procedures can be improved to mitigate the risk of infection with COPs or MDR nosocomial strains. We found increased presence of Gram-negative bacteria following decolonisation treatment, particularly on the skin of individual patients, which could be indicative of a reduced colonisation resistance (Figure 1E). This should be considered a potential risk factor as antibiotic usage at health-care facilities favours and selects for MDR bacterial lineages which complicates treatment considerably [38,39]. High-resolution amplicon-based sequencing approaches and culture-based methods should ideally be used in combination to provide information on how bacterial communities respond to different treatments, along with information on MDR profiles and WGS data on colonising strains. This combinatorial approach can help guide the use of more suitable decolonisation strategies and hopefully reduce the rate of postoperative infections in the future.

## Figures and Tables

**Figure 1 microorganisms-08-01977-f001:**
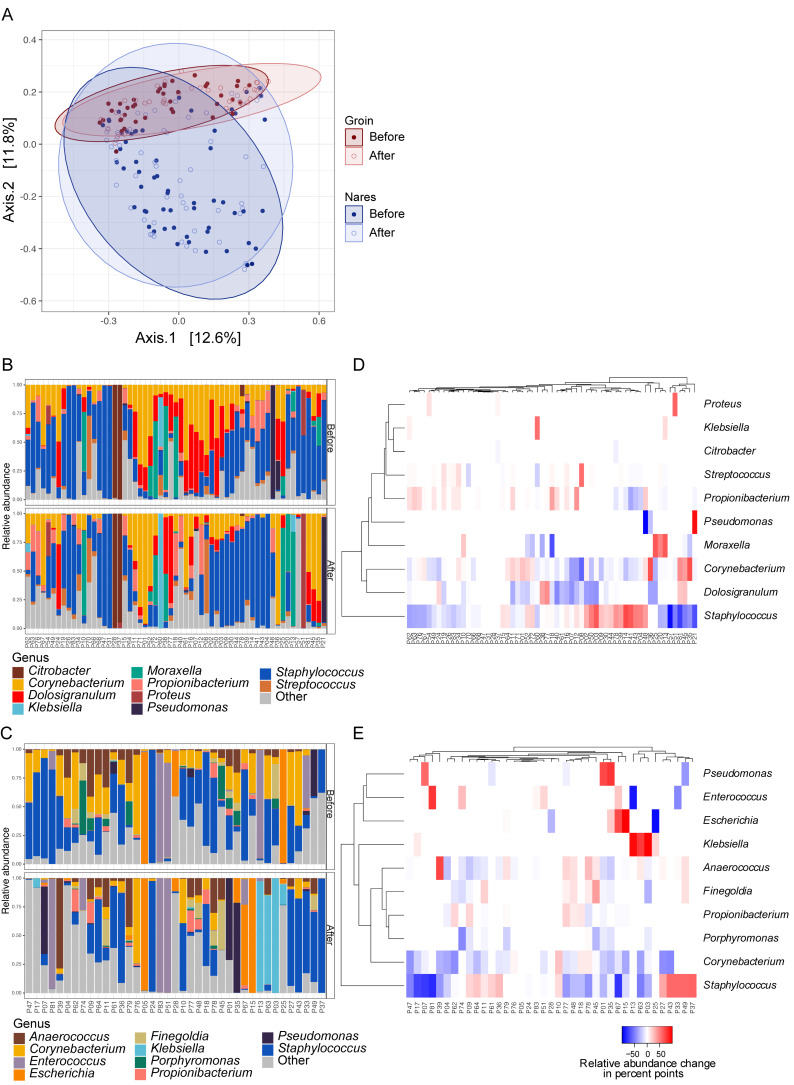
Bacterial community alterations before and after arthroplasty surgery assessed by 16S rRNA (V3-V4) gene sequencing. (**A**) PCoA plot of Hellinger transformed count data based on Bray–Curtis dissimilarity measures. (**B**) Bar plots of 59 bacterial communities from nares and (**C**) 39 groin communities before and after arthroplasty. Heatmaps showing patient-wise relative alterations in percentage points of the 10 most abundant bacterial genera from (**D**) nares and (**E**) groin before compared to after arthroplasty. Patients are clustered according to their similarity in relative abundance alteration for the top 10 most abundant genera. For information on prevalence data for selected bacterial species detected in this study, please refer to Appendix A.

**Figure 2 microorganisms-08-01977-f002:**
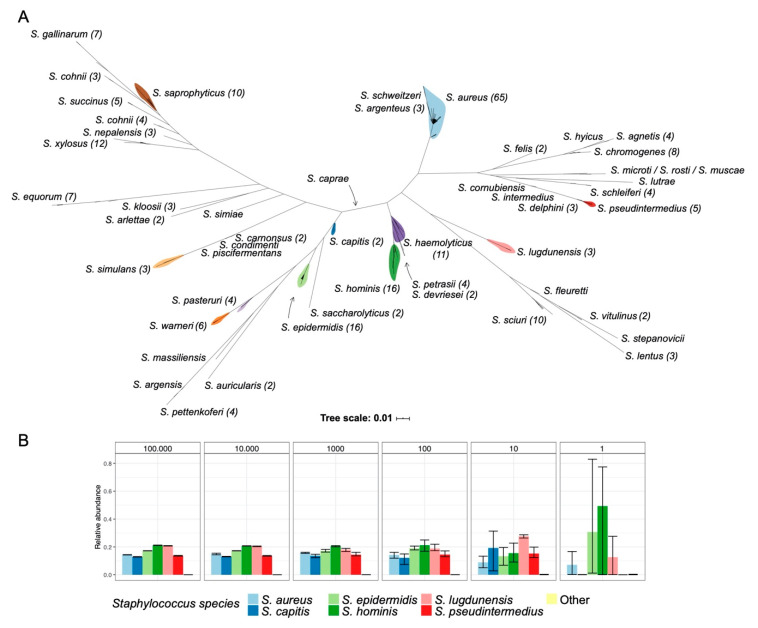
Phylogeny of the *tuf* gene region used in this study. (**A**) Phylogeny based on *tuf* genes from >9000 staphylococcal genomes with *tuf* gene ASVs in each staphylococcal species shown in parenthesis. (**B**) Amplicon sequencing of *tuf* genes from a staphylococcal mock community composed of equal amounts of genomic DNA from *Staphylococcus aureus, Staphylococcus capitis, Staphylococcus epidermidis, Staphylococcus hominis, Staphylococcus lugdunensis,* and *Staphylococcus pseudintermedius* in ≈100,000 copies/organism (left) diluted down to ≈1 copy/organism (right). Error bars indicate highest and lowest observation in each technical triplicate experiment.

**Figure 3 microorganisms-08-01977-f003:**
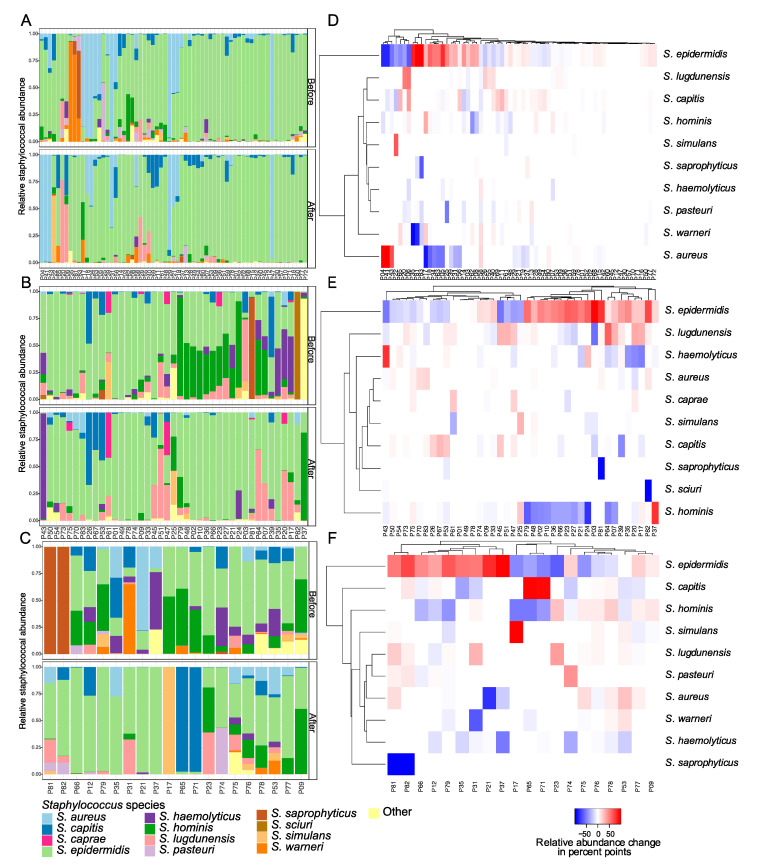
Staphylococcal communities from arthroplasty patients sampled before and after decolonisation treatment and arthroplasty surgery. Bar plots showing alterations in the 10 most abundant *Staphylococcus* species present in (**A**) nares of 65 patients, (**B**) groin of 41 patients, and (**C**) operation site of 20 patients, assessed by *tuf* gene sequencing. Heatmaps show patient-wise relative alterations in percentage points of the 10 most abundant *Staphylococcus* species from (**D**) nares, (**E**) groin, and (**F**) operation site from before compared to after decolonisation treatment and arthroplasty surgery. Patients are clustered according to similarities in relative abundance alterations. For information on prevalence data for all *Staphylococcus* species detected in this study, please refer to Appendix A.

## Data Availability

All sequences are available through the European Nucleotide Archive (ENA) at the European Bioinformatics Institute (EBI) (PRJEB39833). Scripts, phyloseq objects, and the staphylococcal *tuf* gene database are available at https://github.com/ssi-dk/staphylome/.

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
