# Peer review of "Alteration of Bacterial Communities in Anterior Nares and Skin Sites of Patients Undergoing Arthroplasty Surgery: Analysis by 16S rRNA and Staphylococcal-Specific tuf Gene Sequencing"

_microorganisms, 2020, doi:10.3390/microorganisms8121977_

Round 1

Reviewer 1 Report

The authors describe the alterations of bacterial communities in patients undergoing arthroplasties before and after the administration of chlorhexidine gluconate soap decolonization and systemic antibiotic prophylaxis. Those studies are highly warranted and needed to investigate possible negative effects of the so-called horizontal approach (chlorhexidine bath) and prophylactic administration of antibiotic compounds prior surgery. The technical aspects of the study are well-done. The description of the study population should be improved and the discussion should be broadened.

Major comments

  1. The advantage of using tuf instead of other genes should be given for the readership.
  2. In Fig. 3, only the 10 most frequently found staphylococcal species are given. Since in the past years, a lot of newly recognized staphylococcal species have been described, it would be of interest to see also the data of the less frequently detected staphylococcal species (e.g. as suppl. table)
  3. Culturomic-based analyses of the human nose habitats have shown that the fingerprint patterns are highly individually (see e.g. Kaspar et al. Environmental Microbiology (2016) 18(7)), this should be discussed. Moreover, the molecular results by Iversen et al. should be compared with those of a recent culturomic approach in particular with respect to the staphylococcal species found in both approaches.
  4. Since it has been shown that age is a strong factor for accounting for differing bacterial assemblages of anterior nare bacterial communities (see e.g. Camarinha-Silva et al. (2014) Environ Microbiol 16(9): 2939-2952), some details on the patients included should be given (in particular, age and gender distribution, administration of antibiotics). This and putative other influencing factors should be discussed in more detail.
  5. Using nasal swabs, a contamination with the surrounding skin parts are easily possible (see nasal surgical approach as described by Kaspar et al. Environmental Microbiology (2016) 18(7)). While a surgical approach is only appropriate in exceptional cases, this should be mentioned as limitation of the study. In general, I miss a summarized discussion of the limitations of the study.
  6. Fig. 3 A-C, details in these subfigures are very hard to recognize.

Reviewer 2 Report

In this manuscript changes in different microbiomes (nares, groin, surgery site) are investigated before and after arthroplasty surgery. Due to washing with disinfecting soap, antibiotics treatment and careful disinfection of the site of surgery a clear impact on the microbiomes could be expected. Beside the overall composition of the bacterial genera by 16S rRNA sequencing, the authors laid a special focus on the species composition of the genus Staphylococcus, which was analyzed by amplification of the tuf gene, which gives high resolution on species level. interestingly, no dramatic changes in the microbiomes were detected except a reduction in the alpha diversity and a higher sensitivity of some species against the treatment. This manuscript by Iversen et al. is well written and clearly presented. The Supplemental information is comprehensive.

Minor changes

There are some typos in

Line 8: there is a superscript “a” after the last author’s name without explanation. ^

Line 88: delete “Materials and Methods”

Line 212, Figure 2A: there are two “S. cohnii” branches in the figure. Is this an error? Otherwise explain the two positions in the tree.

Line 252: joint instead of join

Supplementary Figure S3: Asterisks instead of Asterix

Supplementary Figure S4: Staphylococcus pseudintermedius instead of Staphylococcus pseudointermedius (no. 29 and 30) and Streptococcus haemolyticus (no. 51) Staphylococcus hyicus subsp. chromogenes (no. 28), Staphylococcus cohnii subsp. urealyticus (no. 13), Staphylococcus nonhaemolytic (no. 32) is not a species name

In the discussion or the results section, the authors should provide some information if any infection after surgery occurred, which might have had an influence on the microbiome composition.

Reviewer 3 Report

A well written paper and timely. This can be of practical use for practitioners in the management of these patients going for elective procedures.

There are improvements that can be made and they are listed below.

In Figure 1 A.

Make the contrasting colors more distinct so that the reader can clearly distinguish groin before from after and nares before from after. For example, might do solid blue circle and dotted blue circle-without fill. Then make the other area solid red circle and dotted red circle without fill. The shades are too similar at this time and make difficulty with seeing the contrast.
